# Topical MTH1 Inhibition Suppresses SKP2-WNT5a-Driven Psoriatic Hyperproliferation

**DOI:** 10.3390/ijms26157174

**Published:** 2025-07-25

**Authors:** Cecilia Bivik Eding, Ines Köhler, Lavanya Moparthi, Florence Sjögren, Blanka Andersson, Debojyoti Das, Deepti Verma, Martin Scobie, Ulrika Warpman Berglund, Charlotta Enerbäck

**Affiliations:** 1Ingrid Asp Psoriasis Research Center, Department of Biomedical and Clinical Sciences, Linköping University, 581 85 Linköping, Sweden; cecilia.bivik.eding@liu.se (C.B.E.); ines.kohler@liu.se (I.K.); lavanya.moparthi@liu.se (L.M.); florence.sjogren@liu.se (F.S.); blanka.andersson@liu.se (B.A.); debojyoti.das@liu.se (D.D.); deepti.verma@liu.se (D.V.); 2Clinical Genomics Linköping, SciLife Laboratory, Department of Biomedical and Clinical Sciences, Linköping University, 581 85 Linköping, Sweden; 3Oxcia AB, 113 34 Stockholm, Sweden; martin.scobie@oxcia.se (M.S.); ulrika.warpmanberglund@oxcia.se (U.W.B.); 4Department of Oncology and Pathology, Karolinska Institute, 171 77 Stockholm, Sweden

**Keywords:** psoriasis, MTH1, keratinocytes, skin, inflammation, proliferation

## Abstract

Topically applied TH1579 alleviated the psoriatic phenotype in the imiquimod-induced psoriasis mouse model by decreasing CD45^+^, Ly6b^+^, and CD3^+^ cell infiltration and downregulating the expression of the proliferation marker PCNA. Moreover, TH1579 strongly suppressed IL-17 expression in mouse skin, accompanied by reduced infiltration of IL-17-producing γδ-T cells. Furthermore, TH1579 decreased keratinocyte viability and proliferation. Mass spectrometry data analysis revealed the enrichment of proteins associated with nucleotide excision repair and cell cycle regulation. The key cell cycle regulatory protein F-box protein S-phase kinase-associated protein 2 (SKP2) was significantly downregulated, along with the psoriasis-associated proliferation marker WNT5a, identified as a SKP2 downstream target. The downregulation of SKP2 and WNT5a was confirmed in MTH1i-treated mouse skin. Our findings support the topical administration of MTH1i TH1579 as a psoriasis treatment. The therapeutic effects depended on the SKP2/WNT5a pathway, which mediates psoriatic hyperproliferation. This study introduces a conceptually innovative topical treatment for psoriasis patients with mild-to-moderate disease who have limited therapeutic options.

## 1. Introduction

An imbalance of reactive oxygen species (ROS) production and the antioxidant defense system is strongly associated with various pathophysiological conditions, including immunological disorders, obesity, arteriosclerosis, and cancer [1]. Specifically, levels of ROS are elevated in cancer cells due to increased proliferation and metabolic rate, which in turn activates their antioxidant defense system. Targeting this protective mechanism has emerged as a promising therapeutic strategy for cancer treatment [2]. The Human MutT Homolog 1 (MTH1, Nudix hydrolase 1, NUDT1) protein is overexpressed in various cancer types [3]. It plays a protective role by eliminating oxidized dNTPs, thereby preventing their incorporation into genomic DNA and maintaining genomic integrity. Consequently, MTH1 inhibition leads to base mispairing, DNA damage, and cell death. Small-molecule inhibitors targeting MTH1 have been developed, and one of them, TH1579 (Karonudib), has demonstrated a favorable safety profile in a first-in-human phase 1 study in patients with advanced solid cancers [4]. Additionally, TH1579 is suggested to have anti-inflammatory properties due to its selective targeting of activated T cells [5].

Psoriasis is a chronic, immune-mediated skin disease characterized by keratinocyte (KC) hyperproliferation and immune cell infiltration, affecting 2–3% of the global population [6].

Psoriasis shares several characteristics with cancer, including excessive proliferation, inflammation, and metabolic disturbances [7]. Increased ROS production in the skin and an impaired cellular antioxidant system play a critical role in the pathogenesis of psoriasis [8,9]. We previously observed the upregulation of MTH1 in the lesional epidermis of human psoriatic skin and subsequently explored the potential of systemic MTH1 inhibition via subcutaneous injection in the imiquimod (IMQ) psoriasis mouse model [10]. MTH1 inhibition normalized the psoriatic phenotype by reducing epidermal cell proliferation and inflammation, thus supporting MTH1 inhibition as a novel promising treatment modality for psoriasis [10].

The majority of psoriasis patients suffer from mild to moderate disease, with topical treatments serving as the primary therapy. However, current topical therapies remain limited, and many patients fail to achieve adequate disease control.

Here, we evaluate the potential of the small-molecule mitotic MTH1 inhibitor (MTH1i) TH1579 (Karonudib), which targets redox imbalance, as a topical treatment candidate for psoriasis. Using the IMQ mouse model, we demonstrate that the short-term application of TH1579 had a significant therapeutic effect, which was dependent on the SKP2/WNT5a pathway, a critical contributor to psoriatic hyperproliferation.

## 2. Results

### 2.1. Topical Treatment with MTH1i TH1579 Alleviates the Psoriatic Phenotype in a Psoriasis Mouse Model

Since TH1579 is a small-molecule inhibitor suitable for topical application, we evaluated its local effects using the IMQ mouse model. The TH1579 concentration in the MTH1i-treated control skin was found to be 202 ± 60 nM, and the plasma concentration was 35 ± 14 nM as analyzed 1 h after administration on day five.

Hematoxylin and eosin (H&E) staining of mouse skin treated with topical IMQ application revealed typical histopathological characteristics of psoriatic inflammation, such as acanthosis, parakeratosis, dermal cell infiltration, and vascular dilation. All these histopathological features were reduced in the skin of IMQ-treated mice following TH1579 administration (Figure 1a).

Immunofluorescence staining revealed a dermal accumulation of CD45^+^ hematopoietic cells in IMQ-treated mouse skin compared to control skin, which was significantly reduced by TH1579 treatment (Figure 1b). Positive staining for the neutrophil marker Ly6b was observed in the dermal cell infiltrate of IMQ-treated mice, while staining was diminished by TH1579 treatment. In addition, anti-CD3 staining in IMQ-treated skin was significantly reduced by the MTH1i cream. Immunofluorescence staining using the proliferating cell nuclear antigen (PCNA) showed pronounced staining in the epidermis of IMQ-treated mice, which was reduced in the TH1579-treated mice and nearly absent in the negative control mice. Thus, topical administration of TH1579 led to a reduction in inflammatory cells and decreased epidermal proliferation.

### 2.2. MTH1 Inhibition Diminishes γδ T Cell Populations in Mouse Skin and Decreases IL-17 and TNF-α Gene Expression

Considering the important role of T cells in driving psoriatic inflammation, we utilized flow cytometry to further characterize the CD3^+^ component of the psoriatic infiltration in the mouse skin. We found a significant decrease in CD3^+^ cells, specifically in IL-17-producing gamma delta (γδ) T cells in the skin of TH1579-treated mice compared to IMQ-only-treated mice (Figure 1c).

To explore the association between the diminished γδ T cell population in mouse skin and the key drivers of psoriatic inflammation, TNF-α and IL-17, we performed mRNA analyses. The gene expression of *TNF-α* was significantly reduced, and the expression of *IL-17* was almost completely abolished by MTH1 inhibition compared to the IMQ-only-treated mouse skin (Figure 1d). Thus, topical administration of TH1579 led to the strong suppression of *IL-17*, which correlated with a reduction in IL-17-producing γδ T cells.

### 2.3. Topical Application of the MTH1i Cream Led to Limited Systemic Effects

The topical application of IMQ has been shown to induce splenomegaly and cause systemic effects on the cellular composition of the spleen, resulting in a shift from lymphoid to myeloid cells [11]. We previously demonstrated reduced splenomegaly and normalized levels of splenic lymphoid cells after subcutaneous injections of MTH1i, indicating systemic effects [10]. In contrast, flow cytometric analysis of the spleen after topical application with MTH1i did not reveal significant changes in the lymphoid (CD3^+^) or the myeloid (Ly6G^+^) cell populations, indicating that the topical treatment had primarily local effects (Appendix A).

### 2.4. MTH1 Inhibition Decreases Keratinocyte Viability and Proliferation

Since hyperproliferation is a hallmark of the psoriatic epidermis, we evaluated the antiproliferative effects of the mitotic MTH1i TH1579 in human epidermal keratinocytes (KCs). Cells were cultured in the presence of 0.01–5 µM MTH1i for 72 h, and the proliferation was assessed using a PrestoBlue cell assay. Cells cultured in concentrations of 0.5 µM and higher demonstrated significantly decreased proliferation compared to untreated control cells (Figure 2a). Consistent with this finding, TH1579-treated KCs (0.5 µM) demonstrated markedly downregulated gene expression of the proliferation marker PCNA (Figure 2b). Furthermore, the antiproliferative effect was confirmed using crystal violet staining of MTH1-inhibited cultures (Figure 2c).

### 2.5. MTH1 Inhibition Downregulates the Cell Cycle Process in Human KCs

To gain insights into the underlying mechanism of TH1579 treatment (0.5 µM) in KCs, we performed unbiased mass spectrometry-based proteomics to determine the alterations in protein profiling (Figure 3a). We first identified proteins with significantly altered expression upon TH1579 treatment by comparing their abundance in the MTH1i-treated samples to that of the vehicle controls. Proteins were considered differentially expressed with a log2 fold change > ±0.58 and FDR-adjusted *p*-value < 0.05. Overall, we identified 2692 significantly differentially expressed proteins (DEPs) after treatment with MTH1i relative to vehicle treatment, of which 2507 were downregulated and 185 were upregulated (Figure 3b). In addition, a heatmap of unsupervised cluster analysis of log2 protein intensities of the DEPs indicates that the majority of these were downregulated upon treatment with MTH1i across the replicates compared to the vehicle controls (Figure 3c). We then performed functional analysis of differentially expressed proteins to identify molecular pathways and proteins associated with MTH1i treatment. Gene Ontology (GO) analysis of biological processes revealed a significant enrichment of proteins involved in chromosome organization, the mitotic cell process, the protein–DNA complex, chromatin organization, and DNA repair (Figure 3d). Moreover, Reactome pathway analysis identified enriched proteins primarily associated with the cell cycle (Figure 3e). This is consistent with the GO, Reactome, and Kyoto Encyclopedia of Genes and Genomes (KEGG) pathway analysis, which revealed the cell cycle, DNA replication, base excision repair, and nucleotide excision repair as the major pathways affected by treatment with MTH1i (Figure 3f). The chord plot represents the relation between the three selected KEGG terms, cell cycle, base excision repair, and nucleotide excision repair, and their associated proteins (Figure 3g). Collectively, these findings suggest that TH1579 treatment predominantly leads to the downregulation of cell cycle regulatory proteins and DNA repair mechanisms in KCs.

### 2.6. MTH1 Inhibition Interferes with the SKP2/WNT5a Pathway

Our mass spectrometry results demonstrated that TH1579 treatment led to changes in the expression of cell cycle-regulating proteins. Particularly, we found a significant downregulation of F-box protein S-phase kinase-associated protein 2 (SKP2) (Figure 4a), which is associated with the ubiquitin-mediated degradation of cell cycle components and target proteins involved in cell cycle progression [12,13]. When further elucidating downstream effectors of SKP2, we identified that the proliferation marker PCNA and the key cell cycle regulator cyclin-dependent kinase 4 (CDK4) exhibited markedly reduced protein expression following MTH1 inhibition. Moreover, the CDK4/6 cell cycle inhibitor CDKN1A (p21), which is proteolyzed by SKP2 through ubiquitin-mediated degradation, showed a tendency toward upregulation at the protein level. These findings were further validated by qPCR, which revealed a markedly significant decrease in the mRNA levels of *SKP2*, *CDK4*, and *PCNA*, along with an increase in *CDKN1A* expression upon TH1579 treatment (Figure 4b). Thus, we demonstrate an antiproliferative effect of MTH1 inhibition mediated by the SKP2 proliferation pathway.

We next investigated the effect of MTH1i on cellular markers previously implicated in KC proliferation. WNT5a (Wnt Family Member 5a) was previously reported to promote KC proliferation and malignant progression [14,15]. We found that *WNT5a* was significantly increased by IL-17 and TNFα treatment in KCs, supporting the association with the psoriatic phenotype (Figure 4c). KCs treated with TH1579 exhibited significantly decreased *WNT5a* gene expression (Figure 4c), which was also observed in mass spectrometry data (Figure 4d). These results indicate that the antiproliferative effects of MTH1 inhibition involve WNT5a signaling.

### 2.7. WNT5a Acts Downstream of SKP2 and Is Dependent on MTH1

To assess whether WNT5a is a downstream target of SKP2, KCs were transfected with SKP2 siRNA, and the expression levels of the downstream genes *CDKN1A* and *WNT5a* were analyzed. *CDKN1A* showed increased expression following successful SKP2 downregulation (Figure 5a). In addition, we found that *WNT5a* was significantly downregulated in SKP2-silenced cells. These findings suggest that WNT5a is a SKP2 downstream target.

To confirm that the observed effects in MTH1i-treated cells were specifically MTH1-dependent, and not due to off-target effects of TH1579, KCs were transfected with MTH1-targeting siRNA. *MTH1* gene expression was successfully reduced to 10% of its baseline level (Figure 5b). In MTH1 silenced cells, the mRNA expression of *SKP2*, *CDK4*, *WNT5a*, and *PCNA* was significantly reduced, while *CDKN1A* was upregulated, closely reflecting the changes observed following treatment with the TH1579 (Figure 5c). These findings confirm that the effects of TH1579 are indeed dependent on MTH1; however, off-target effects cannot be excluded.

### 2.8. MTH1i Treatment Results in Downregulation of Skp2 and Wnt5a in Mice

Our in vitro analyses suggest that MTH1i acts through SKP2/WNT5a. We found that the skin of IMQ-treated mice exhibited increased Skp2 and Wnt5a protein expression compared to control skin (Figure 6). MTH1i application resulted in significantly reduced staining of both Skp2 and Wnt5a. These in vivo findings validate the in vitro results, demonstrating that the antiproliferative effect of the inhibitor is mediated by the SKP2-WNT5a proliferation pathway.

## 3. Discussion

MTH1 activity is essential for cellular survival under conditions of increased oxidative stress [3]. The inhibition of MTH1 results in oxidized dGTPs being incorporated into DNA during mitotic replication, which induces cell death. The lack of protection from oxidative damage by MTH1 inhibition is currently being explored as a new cancer treatment strategy in solid tumors as well as hematological cancers [4,16]. In this study, a topical formulation of MTH1i TH1579 was evaluated for the treatment of psoriasis, a common hyperproliferative and inflammatory skin disease.

Using the IMQ psoriasis mouse model, we found that the typical histopathological characteristics of psoriatic inflammation were distinctly reduced in the skin of TH1579-treated mice, along with a decrease in inflammatory infiltrates. Mouse skin contains CD3^+^ dermal T cells, including resident CD3^+^ γδ T cells, which are considered the primary producers of IL-17 upon IL-23 and IL-1β stimulation [17]. Recently, Karsten et al. demonstrated that TH1579 selectively depletes activated human T cells during increased oxidative stress in vitro [5]. Chen et al. reported a decreased overall frequency of activated CD3^+^ T cells following MTH1i treatment in a mouse hepatitis model [18]. Consistently, our study shows that MTH1i significantly reduced the frequency of CD3^+^ T cells in the mouse skin, particularly γδ T cells, which likely accounts for the decreased *IL-17* gene expression observed. As IL-17-targeting monoclonal antibodies are effective as systemic psoriasis therapy for patients with moderate to severe psoriasis [19,20], our findings suggest that topical MTH1 inhibition could serve as a potentially effective local treatment by reducing IL-17 levels in the skin.

To gain comprehensive insight into the effects of TH1579 in human KCs, we used mass spectrometry-based proteomics. Our data demonstrated the differential expression of components of the base excision repair pathway in TH1579-treated KCs, which confirms the established function of these mitotic MTH1 inhibitors. Our data further highlighted pathways involved in cell cycle regulation. These findings were further supported by the reduced PCNA expression in KCs both in vitro and in vivo after MTH1 inhibition. Interestingly, we found a strong downregulation of the proliferation-associated protein SKP2 compared to control skin following MTH1 inhibition. SKP2 is frequently upregulated in various types of cancers and is associated with cancer progression and metastasis [21,22]. Recent studies have also reported increased SKP2 expression in the lesional skin of psoriasis patients, as well as in the IMQ mouse model [23]. In addition, SKP2 has been shown to be implicated in psoriatic proliferation and angiogenesis [23,24]. Its role in promoting cell cycle progression involves targeting CDK inhibitors, including p27, p21, and p57 for K48-linked ubiquitination and degradation [12,13,25], but may also involve transcriptional regulation [12]. We demonstrated that MTH1 inhibition led to a significant upregulation of p21 (CDKN1A), which plays a critical role in halting cell cycle progression. This result is consistent with previous studies conducted on colon cancer xenograft mice treated with TH1579 [26] and on U2OS cells treated with TH287 and TH588 [3], resulting in tumor growth inhibition and reduced cell proliferation. SKP2 knockdown by siRNA led to an increase in CDKN1A mRNA levels, suggesting that SKP2 may be involved in the direct or indirect regulation of CDKN1A transcription.

Interestingly, Wang et al. reported a link between SKP2 and MTH1 in melanoma cells, where SKP2 physically binds to and stabilizes MTH1 via K63-linked polyubiquitination, thus sustaining its expression [25]. However, they did not observe any change in SKP2 levels following MTH1 silencing, which contrasts with our findings. This discrepancy may be attributed to differences in cell type. Our finding of SKP2 downregulation may be a direct or indirect effect of MTH1 inhibition. The expression of SKP2 is primarily regulated at the transcriptional level, with numerous transcription factors associated with its promoter region. Additionally, SKP2 expression is influenced by several signaling pathways, such as Notch1, PI3K/AKT, and IL-6/JAK2/STAT3 [27]. Notably, MTH1 has previously been shown to regulate the PI3K/AKT pathway in gastric cancer [28] and the STAT3 pathway in breast cancer [29].

Wnt signaling has been shown to play important roles in regulating cell proliferation, differentiation, and inflammation [15,30]. WNT5a was reported to be several-fold upregulated in psoriasis skin lesions [31,32] and has been identified as one of five key genes associated with psoriasis pathogenesis [33]. The knockdown of WNT5a led to suppressed cell proliferation and induced apoptosis in HaCaT cells and normal human KCs [34]. We previously found that components of the Wnt signaling pathway, and specifically *WNT5a*, were strongly differentially hypomethylated in the epidermis of psoriasis patients compared to controls [35].

When using SKP2 siRNA transfection, we identified WNT5a as a downstream target of SKP2, suggesting that the MTH1 inhibition exerts anti-proliferative effects through the suppression of SKP2-WNT5a signaling.

In summary, we demonstrate significant improvements in the IMQ mouse model following short-term therapy with topically applied MTH1i TH1579. In addition to the established effect of MTH1 inhibition in counteracting oxidative stress, we demonstrate that MTH1 inhibition prevents KC proliferation by downregulating SKP2 and WNT5a signaling. Our findings provide insights into the molecular pathways targeted by MTH1 inhibition and its efficacy in topical application, offering a promising approach for future psoriasis treatments.

## 4. Materials and Methods

### 4.1. Cells Culture Condition

Human epidermal neonatal KCs, HEKn, were cultured in complete EpiLife medium supplemented with 1% EpiLife defined growth supplement (EDGS) (all from Cascade Biologics, Portland, OR, USA), 1% amphotericin B (Gibco, ThermoFisher Scientific, Waltham, MA, USA), and 1% penicillin/streptomycin (Gibco).

### 4.2. Mouse Experiments

C57B6/J female mice were obtained from The Jackson Laboratory (Bar Harbor, ME, USA). For the IMQ-treated mice, 62.5 mg of Aldara cream (5% imiquimod, Meda AB, Solna, Sweden) was applied to a shaved area of the back (3 × 2 cm) for four consecutive days to induce psoriasis-like inflammation. MTH1i (TH1579, 30 mg/mL formulated in PEG300 and Lanogen mixture) or vehicle cream was applied to these mice 4 h after the IMQ cream application for three days (days 2–4). Control mice were sham-treated with only Vaseline cream (ACO, Kista, Sweden) or in combination with vehicle cream. After five days, the mice were sacrificed, and their skin and spleens were collected.

### 4.3. RNA Extraction, cDNA Synthesis, and qPCR Analysis

The mouse skin was harvested and snap-frozen in liquid nitrogen. Prior to RNA extraction, the tissue was disrupted and homogenized using TissueLyser (Qiagen, Hilden, Germany). Total RNA from mouse skin and KCs was isolated using RNeasy kits for fibrous tissue and RNeasy Plus kit (Qiagen), respectively, according to the manufacturer’s instructions. For analyzing the gene expression, cDNA synthesis was performed using the Maxima First Strand cDNA Synthesis Kit (ThermoFisher Scientific). qPCR, using the TaqMan Fast Advanced Master Mix with pre-designed TaqMan Gene expression assays (Appendix A), was performed on a QuantStudio 7 flex system (Applied Biosystems, Foster City, CA, USA) and samples were run in triplicates. The results were normalized to the housekeeping gene *RPLP0* using the comparative Ct (2^−∆∆Ct^) method.

### 4.4. Immunofluorescence and H&E-Staining

Mouse skin sections were fixed in 4% formaldehyde, paraffin-embedded, and cut at 6–7 μm. Histolab-clear (Histolab Products, Gothenburg, Sweden) was used for deparaffinization and ethanol for rehydration. After incubation with Citrate antigen retrieval buffer (pH = 6, Invitrogen, ThermoFisher Scientific) at 97 °C and blocking with 5% bovine serum albumin for 45 min at room temperature, primary antibodies (Appendix A) were applied overnight at 4 °C. Alexa Fluor 555 plus for one hour was used as secondary antibody parallel with DAPI (4′,6-diamidino-2-phenylindole) to counterstain the nuclei. Negative controls were attained using isotype control IgG antibody (Novus Biologicals, Littleton, CO, USA) or by exclusion of the primary antibody. Some sections were stained with hematoxylin and eosin (H&E, Histolab Products).

The immunofluorescence sections were examined in a Leica DMi8 inverted fluorescent microscope (Leica Microsystems GmbH, Wetzlar, Germany), and fluorescence intensities were measured using the ImageJ Fiji 2.15.0 software (National Institutes of Health, Bethesda, MD, USA). The H&E-stained sections were examined under an Olympus BX 51 microscope (Olympus Corporation, Tokyo, Japan).

### 4.5. Statistics

Student’s *t*-test or one/two-way ANOVA test followed by Šídák’s or Dunnett’s multiple comparisons test was used to determine significance. A *p*-value ≤ 0.05 was considered significant. All data are presented as the mean ± standard error of the mean (SEM). GraphPad Prism 10.0.2 (GraphPad Software, La Jolla, CA, USA) was used for statistical analyses.

Detailed information on mass spectrometry analysis, cell viability and proliferation assays, and flow cytometry analysis are provided in Appendix A.

## Figures and Tables

**Figure 1 ijms-26-07174-f001:**
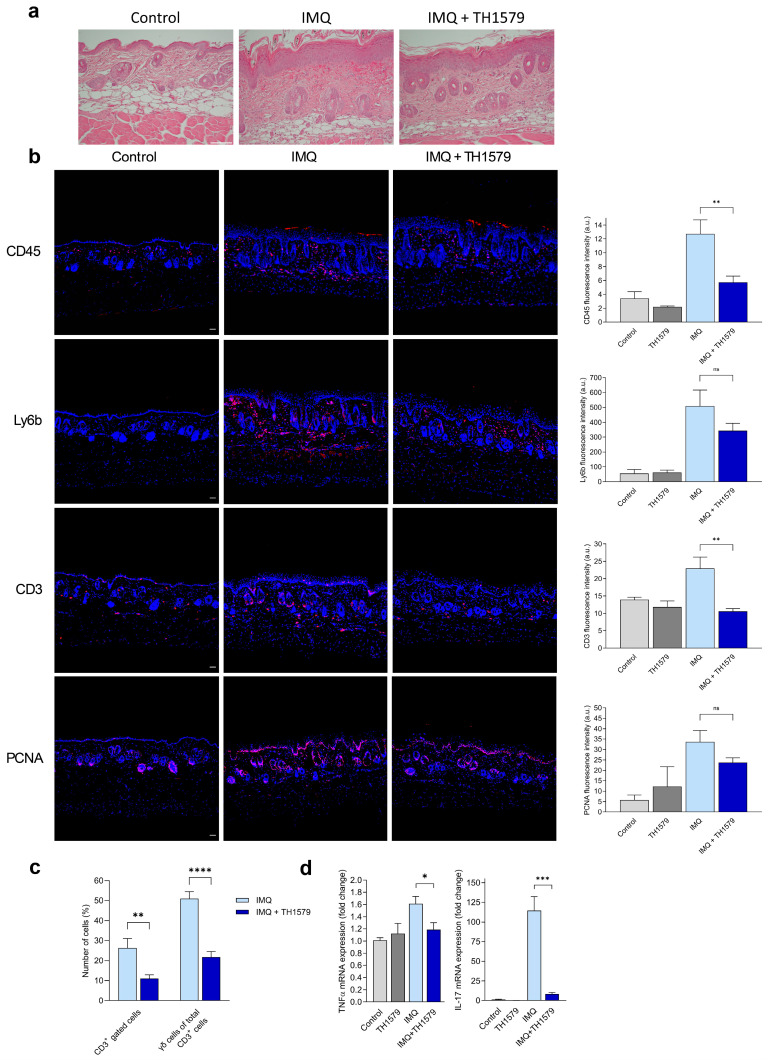
MTH1 inhibition alleviates the psoriatic phenotype and reduces inflammation. Aldara cream (IMQ) was applied to the shaved backs of mice (C57B6/J) for four consecutive days. Four hours after IMQ treatment, the mice underwent daily topical application of the MTH1 inhibitor (TH1579) for 3 days (days 2–4). n = 6. (**a**) Hematoxylin and eosin-stained skin sections of IMQ- and MTH1 inhibitor (TH1579)-treated mice. Bar = 100 µm. (**b**) Immunofluorescence staining (red) of CD45^+^, Ly6B^+^, and CD3^+^ cells and proliferating cell nuclear antigen (PCNA) in mouse skin. Nuclear counterstaining with DAPI (blue). Fluorescence intensity measurements were performed. n = 3–6, bar = 50 µm. (**c**) Flow cytometry analyses of CD3^+^, and γδ T cells in MTH1 inhibitor (TH1579) and IMQ-treated mouse skin (n = 6). (**d**) mRNA expression analyses of *TNFα* and *IL-17* in MTH1 inhibitor (TH1579) and IMQ-treated skin (n = 5–6). * *p* < 0.05, ** *p* < 0.01, *** *p* < 0.001, **** *p* < 0.0001, ns non-significant.

**Figure 2 ijms-26-07174-f002:**
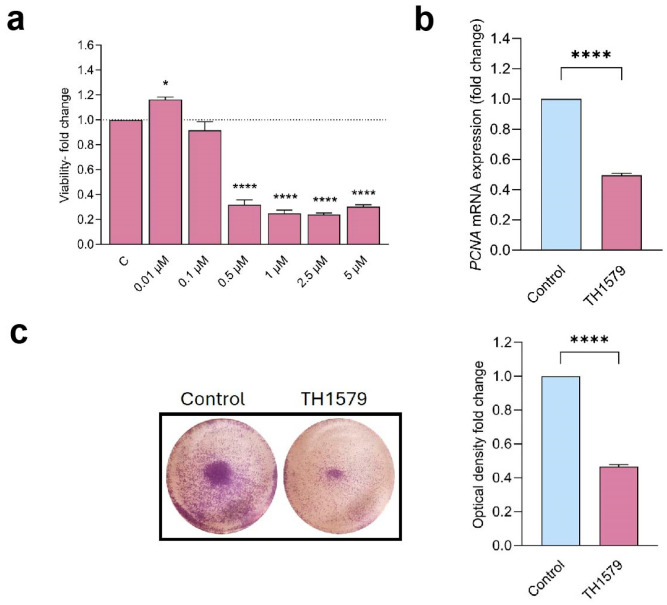
MTH1 inhibition with TH1579 reduces keratinocyte viability and proliferation. (**a**) The viability of HEKn was assessed following treatment with varying concentrations of the MTH1 inhibitor (0.01–5 µM, TH1579) for 72 h (n = 3). (**b**) *PCNA* mRNA expression was quantified 48 h post-treatment (n = 4). (**c**) Crystal violet staining was performed to evaluate cell proliferation after 48 h of TH1579 treatment (n = 3). * *p* < 0.05, **** *p* < 0.0001.

**Figure 3 ijms-26-07174-f003:**
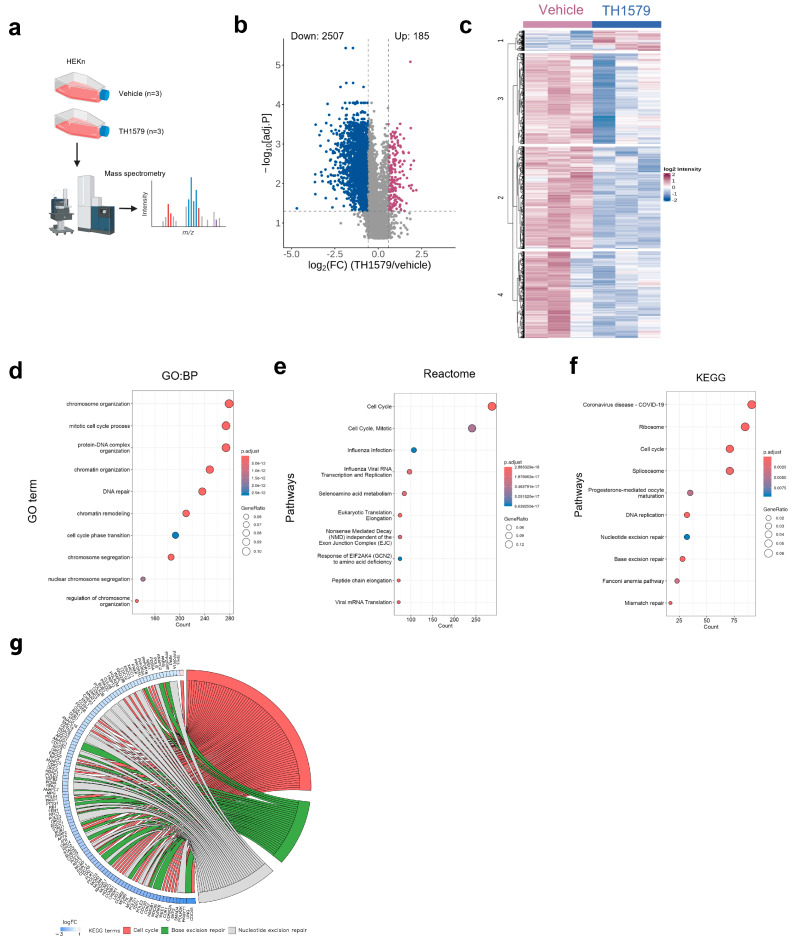
Mass spectrometry-based proteomic analysis of MTH1 inhibitor TH1579, which induces protein changes in human keratinocytes. (**a**) Schematic representation of the mass spectrometry-based proteome profiling workflow. Figure generated by BioRender software, https://app.biorender.com/, accessed on 1 February 2025. (**b**) The volcano plot illustrates differentially expressed proteins (DEPs) between TH1579 treatment vs. control. Significantly DEPs are denoted as colored dots (blue-downregulated, red-upregulated) with FDR-adjusted *p*-value < 0.05 and log2 fold change > ±0.58. (**c**) Heatmap of unsupervised cluster analysis of the log2 protein intensities profiles of 2692 differentially expressed proteins. Dot plots of Gene Ontology (GO) of biological process (**d**), Reactome (**e**) and Kyoto Encyclopedia of Genes and Genomes (KEGG) (**f**). Pathway analysis of DEPs displaying top 10 significantly (Benjamini–Hochberg FDR correction, *p* < 0.05) enriched terms. (**g**) The chord plot shows the three selected significant KEGG pathway enrichments and corresponding differentially expressed proteins connected by chords. Proteins are arranged based on log2 fold change.

**Figure 4 ijms-26-07174-f004:**
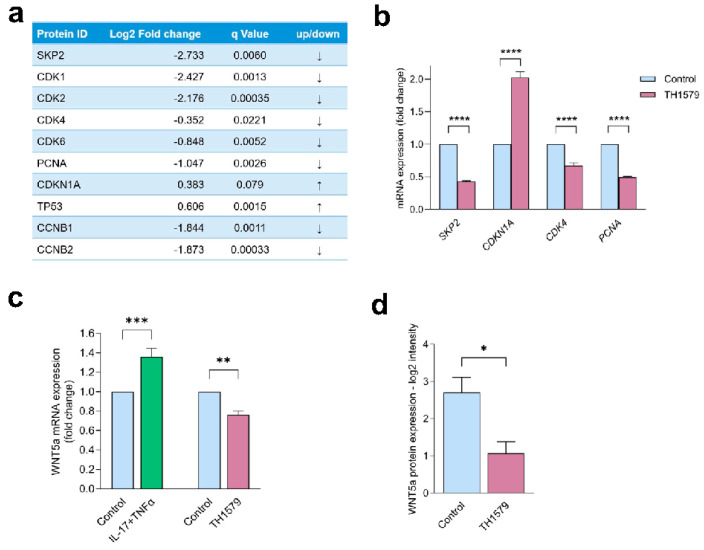
MTH1 inhibition modulates cell cycle regulatory proteins, including SKP2 and WNT5a. (**a**) Protein expression analysis of cell cycle regulating proteins in keratinocytes was determined by mass spectrometry following treatment with the MTH1 inhibitor TH1579 (0.5 µM) (n = 3). (**b**) mRNA expression of the genes *SKP2*, *CDKN1A*, *CDK4*, and *PCNA* was analyzed (n = 4). (**c**) mRNA expression of *WNT5a* was analyzed in HEKn in response to IL-17 and TNFα and the MTH1 inhibitor TH1579 (0.5 µM) treatment (n = 4). (**d**) Protein expression of WNT5a was evaluated using mass spectrometry (n = 3). * *p* < 0.05, ** *p* < 0.01, *** *p* < 0.001, **** *p* < 0.0001.

**Figure 5 ijms-26-07174-f005:**
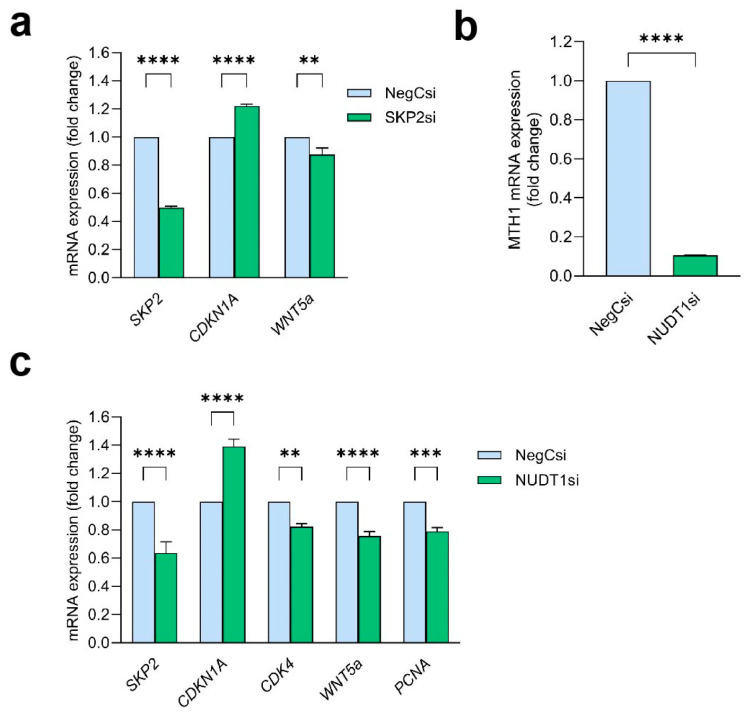
WNT5a is located downstream of SKP2 and is dependent on MTH1. (**a**) The mRNA expression of *SKP2*, *CDKN1A*, and *WNT5a* were determined following siRNA-mediated silencing of SKP2 in HEKn. *G*ene expression was compared with negative control siRNA-transfected cells. mRNA expression of (**b**) MTH1 (*NUDT1*) and (**c**) *SKP2*, *CDKN1A*, *CDK4*, *WNT5a*, and *PCNA* was analyzed in HEKn cells transfected with *NUDT1* siRNA. *G*ene expression was compared to negative control siRNA-transfected cells. n = 4, ** *p* < 0.01, *** *p* < 0.001, **** *p* < 0.0001.

**Figure 6 ijms-26-07174-f006:**
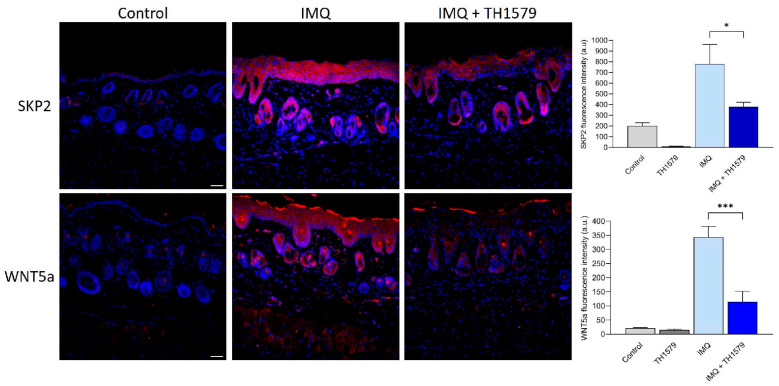
Decreased protein expression of SKP2 and WNT5a in mouse skin treated with MTH1 inhibitor. Aldara cream (IMQ) was applied to the shaved backs of mice (C57B6/J) for four consecutive days. Four hours after IMQ treatment, the mice underwent daily topical application of the MTH1 inhibitor (TH1579) for 3 days (days 2–4). Immunofluorescence staining of SKP2 and WNT5a (red) in mouse skin and nuclear counterstaining with DAPI (blue). Fluorescence intensity measurements were performed. n = 3–6, bar = 50 µm, * *p* < 0.05, *** *p* < 0.001.

## Data Availability

The mass spectrometry proteomics data have been deposited to the ProteomeXchange Consortium via the PRIDE [36] partner repository with the dataset identifier PXD061700.

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
