# Peer review of "Topical MTH1 Inhibition Suppresses SKP2-WNT5a-Driven Psoriatic Hyperproliferation"

_ijms, 2025, doi:10.3390/ijms26157174_

Round 1

Reviewer 1 Report

Comments and Suggestions for Authors

In this manuscript, the authors investigated whether MTH1 inhibitor(MTH1i) TH1579 can be used as a topical treatment for psoriasis. Using an IMQ mouse model, they showed that MTH1i application alleviated the psoriatic phenotype and suppressed immune cell infiltration. They further showed that MTH1i treatment inhibited keratinocyte proliferation and led to changes in the levels of proteins involved in cell cycle, DNA replication, etc. Overall, it is an interesting topic to study with potential therapeutic indications. However, the authors failed to provide sufficient evidence to support their conclusion whereby MTH1i suppresses psoriasis.

  1. The resolution of the figures is not high enough. The pathways and genes highlighted in Figure 3 are too small to read (zooming in did not help due to low resolution).
  2. Figure 1b: Stainings for CD45 and Ly6b are very faint and difficult to see. It would be great to replace them with better images. 
  3. Suppl fig 1: Couldn't open the file. Please export it as jpg or tiff instead.
  4. Figure 4a, b: Since p21 mRNA level increases upon MTH1i treatment, it is hard to conclude whether the higher p21 protein level is due to increase in expression or decrease in proteasomal degradation mediated by SKP2 as suggested in the text. The protein half-life of p21 needs to be measured +/- MTH1i to tease this apart. 
  5. Line 174: A direct SKP2 downstream target indicates a protein whose degradation is directly mediated by SKP2. Since WNT5a is down-regulated upon SKP2 knockdown, this is likely a secondary effect rather than a direct target.
  6. Figure 5c: The fold change in mRNAs upon MTH1 knockdown is much smaller compared to that with inhibitor treatment. Is this due to drug/siRNA efficacy, or does it suggest an off-target effect of the inhibitor? If MTH1-silenced cells are treated with MTH1i, will there be further changes in the mRNA levels of the genes measured here?
  7. Line 188-189: To demonstrate that SKP2 and WNT5a mediate the effect of MTH1i, rescue experiments need to be conducted to determine if SKP2 or WNT5a overexpression can rescue the anti-proliferative effects of MTH1i.
  8. Would it be possible to do mass spec on MTH1i-treated mouse skin tissue? It might provide more physiologically relevant results.
  9. Suppl fig 2 is not mentioned in the text.

Reviewer 2 Report

Comments and Suggestions for Authors

Comment to IJMS manuscript titled " Topical MTH1 inhibition suppresses SKP2-WNT5a-driven psoriatic hyperproliferation”

The manuscript explored the topical use of the MTH1 inhibitor TH1579 (Karonudib) as a potential treatment for psoriasis. The authors demonstrate that TH1579 reduces keratinocyte proliferation and inflammation in both in vitro and in vivo models, with a particular focus on the SKP2/WNT5a signaling pathway. Below is a list of questions regarding this study.

  1. All the images presented in Figure 3 exhibit significant blurriness, which detracts from their clarity and effectiveness.

  1. The authors should discuss whether the downregulation of SKP2 is a direct or indirect consequence of MTH1 inhibition. Please elaborate on the mechanistic connection between MTH1 inhibition and the downregulation of SKP2.

  1. The graphical abstract, while mentioned on line 379, is not included in the PDF. If applicable, please ensure it is provided with the final submission.

Round 2

Reviewer 1 Report

Comments and Suggestions for Authors

The authors have addressed my comments.

It would be helpful if the arrows pointing to WNT5a and p21 were dotted in the graphic abstract, as it is not shown in the manuscript that this is a direct effect.
